# Heritability of Immunity Traits and Resistance of Atlantic Salmon against the Sea Louse *Caligus rogercresseyi*

**DOI:** 10.3390/biology12081078

**Published:** 2023-08-02

**Authors:** Débora Torrealba, Byron Morales-Lange, Victoriano Mulero, Anti Vasemägi, Luis Mercado, José Gallardo-Matus

**Affiliations:** 1Laboratorio de Genética y Genómica Aplicada, Escuela de Ciencias del Mar, Pontificia Universidad Católica de Valparaíso, Avenida Universidad 330, Valparaíso 2373223, Chile; debora.torrealba@pucv.cl; 2Grupo de Marcadores Inmunológicos en Organismos Acuáticos, Instituto de Biología, Pontificia Universidad Católica de Valparaíso, Avenida Universidad 330, Valparaíso 2373223, Chile; byron.morales@pucv.cl (B.M.-L.); luis.mercado@pucv.cl (L.M.); 3Departamento de Biología Celular e Histología, Facultad de Biología, Universidad de Murcia, C. Campus Universitario, 5, 30100 Murcia, Spain; vmulero@um.es; 4Department of Aquatic Resources, Swedish University of Agricultural Sciences. Almas Allé 8, SE-750 07 Uppsala, Sweden; anti.vasemagi@slu.se; 5Institute of Veterinary Medicine and Animal Sciences, Estonian University of Life Sciences, Friedrich Reinhold Kreutzwaldi 1a, 51014 Tartu, Estonia

**Keywords:** Nkef, Tnfα, Il-8, cytokines, sea lice, Atlantic salmon, heritability, innate immunity

## Abstract

**Simple Summary:**

Sea lice are a relevant ectoparasite that affect the salmon industry. Not all fish respond in the same way to this parasite—some fish are resistant and have a low sea lice load, and others are susceptible and have a high parasite load. In this work, we studied the heritability of three key proteins associated with the innate immunity and resistance of Atlantic salmon against sea lice. Nkef, Tnfα, and Il-8 were quantified by indirect ELISA in order to determine their heritability in different families of Atlantic salmon. Our results show that the expression of Nkef and Tnfα proteins are highly heritable and are related to resistance to sea lice in Atlantic salmon.

**Abstract:**

The immune response of Atlantic salmon to sea lice has been extensively studied, but we still do not know the mechanisms by which some fish become resistant and others do not. In this study, we estimated the heritabilities of three key proteins associated with the innate immunity and resistance of *Salmo salar* against the sea louse *Caligus rogercresseyi*. In particular, we quantified the abundance of 2 pro-inflammatory cytokines, Tnfα and Il-8, and an antioxidant enzyme, Nkef, in Atlantic salmon skin and gill tissue from 21 families and 268 individuals by indirect ELISA. This covers a wide parasite load range from low or resistant (mean sea lice ± SE = 8.7 ± 0.9) to high or susceptible (mean sea lice ± SE = 43.3 ± 2.0). Our results showed that susceptible fish had higher levels of Nkef and Tnfα than resistant fish in their gills and skin, although gill Il-8 was higher in resistant fish, while no significant differences were found in the skin. Furthermore, moderate to very high heritable genetic variation was estimated for Nkef (h^2^ skin: 0.96 ± 0.14 and gills: 0.97 ± 0.11) and Tnfα (h^2^ skin: 0.53 ± 0.17 and gills: 0.32 ± 0.14), but not for Il-8 (h^2^ skin: 0.22 ± 0.12 ns and gills: 0.09 ± 0.08 ns). This work provides evidence that Nkef and Tnfα protein expressions are highly heritable and related to resistance against sea lice in Atlantic salmon.

## 1. Introduction

Sea lice are considered one of the most serious problems in aquaculture production [1,2]. These parasites are parasitic copepods of the family Caligidae and mainly represented by two genera, *Caligus* and *Lepeophtheirus*. Two of the principal ectoparasite of these genera affecting salmon farming are *Caligus rogercresseyi* in the southern hemisphere and *Lepeophtheirus salmonids* in the northern hemisphere [3,4]. The harmful effect of these parasites is widely known; they can seriously damage the skin, increase stress, reduce growth, and cause high mortality [5,6,7]. As a consequence of severe mechanical damage, Caligidae parasites evoke a strong innate immune response in the host and also perturb the mucosal microbiome of fish [8,9]. Even though several methods exist (e.g., feed supplements, bath treatments, temperature shock, flushing the surface of the fish with pressurized water, tailored cage design, or “lice-zapping” lasers) [10,11,12,13] to avoid or control this parasite, the results have been restricted. In Chile, *Caligus rogercresseyi* is one of the most relevant sanitary issues in aquaculture with a high prevalence in salmon farms increasing the production cost by 1.4 USD per kilogram of produced salmon [14,15]. Other detrimental effects of *C. rogercresseyi* are that it: (1) reduces the resistance of Atlantic salmon (*Salmo salar*) to other pathogens [16,17], (2) overrides the protective effects of vaccination against *Piscirickettsia salmonis* in Atlantic salmon [17], and (3) potentially acts as a reservoir for fish pathogens, such as *Vibrio*, *Tenacibaculum*, and *Aeromonas*, present in their microbiota [18]. *C. rogercresseyi* has eight developmental stages which include two free-swimming nauplius, a copepodid stage that infests and attaches to the host, four attached chalimus, and finally, a mobile adult stage [19]. Usually, newly attached copepotid stages and chalimus stages are known as sessile parasites in contrast to the adult that is capable of moving on the host.

Sea lice infestation in Atlantic salmon elicits an early upregulation of several immune response components, such as inflammatory response, cytokine production, Tnf and NF-kappa B signaling, protease secretion, and complement activation [20,21,22]. Particularly, resistance to *Caligus* has been potentially associated with an upregulation of immune receptors (TLR and C-X-C chemokine receptor for Il-8), an upregulation of genes related to the TH2 pathway, and the depletion of cellular iron availability [20,21]. Furthermore, Atlantic salmon could modulate sea lice transcriptome at early infestation stages as a defence mechanism. For example, reactive oxygen species (ROS) emitted by infected fish could modulate the sea lice antioxidant system [23].

Fish skin is the first barrier to the exterior environment and plays an essential role in protection against pathogens [24]. In healthy fish, skin immune-related molecules can be found as lectins, cytokines, lysozyme, and complement proteins [24,25]. During a sea lice infestation, copepods attach to the skin or fins and feed on the skin, mucus, and blood, leading to open wounds [20] and producing a modulation of the expression of immune response-related genes, such as cytokines and receptors [9,20,21,26]. This modulation is observed even in the undamaged skin of infested fish [27]. Similar to skin, gills are an external barrier to pathogens that are responsible for gas exchange, osmoregulation, and ion exchange. However, the detection of sea lice in gills is not typical, although the immune response has been detected in this tissue during *C. rogrecresseyi* infestation [27,28]. Sea lice can produce an anti-inflammatory response, hyperplasia, thickening, and complete or partial lamellar fusion in fish gills even when they are not attached on it [29]. Simultaneously, the gills of infected fish show increased melanomacrophages centers/eosinophilic granule cells, monocytes/macrophages, and lymphocytes, with an increase in mucus production and globet cells [29]. Hence, gills are highly sensitive to *Caligus* infestation, being a good candidate for evaluating immune parameters.

In this study, we evaluated three immune-defence related proteins, specifically, natural killer enhancing factor (Nkef), tumor necrosis factor-alpha (Tnfα), and interleukin 8 (Il-8). Nkef, is a member of peroxiredoxin (Prx) family, is an effector molecule that plays a supportive role in cell signaling and immunomodulation in mammals, playing a protective antioxidant activity role, catalyzing the reduction of peroxides avoiding cellular damage [30,31]. In teleost, Nkef is up-regulated in gills and skin and other tissues after a parasitic, bacteria, and viral infection [27,32,33,34,35]. Recombinant Nkef protein has been shown to enhance the cytotoxic capacity of non-specific cytotoxic cells from kidney in Nile tilapia (*Oreochromis niloticus*) [34]. Cell-types producing Nkef are resembling T-cells and macrophages in rainbow trout (*Oncorhynchus mykiss*), red blood, and epithelial cells in common carp (*Cyprinus carpio*) [27,35]. Furthermore, we have previously demonstrated at the protein level that Nkef abundance increases in Atlantic salmon gills and skin challenged with *C. rogercresseyi* [27].

Cytokines are regulators of the homeostasis of host defences, determining the type of response generated and the effector mechanisms generated to mediate resistance. We focused our work in two cytokines Tnfα and Il-8, related to Th1 response. Tumour necrosis factor alpha is a cytokine mainly secreted by activated monocytes and macrophages [36,37]. Tnfα has a dual role, first as pro-inflammatory mediator when it is secreted by classically activated macrophages [36,37,38,39,40], and second as an initiator of tissue repair and antioxidant defence when produced by alternatively activated macrophages [41]. Additionally, fish Tnfα is suggested to be involved in regulating leukocyte homing, proliferation, and migration [42]. As the third host immune response molecule, we quantified the protein abundance of Il-8, a potent cytokine mediator; its primary role is to recruit and activate neutrophils and induce the migration of monocytes, lymphocytes, and other granulocytes subsets to the infection site [43,44]. In pink salmon, *Oncorhynchus gorbuscha*, an early and high gene expression of Il-8 in fin has been suggested as a molecular mechanism behind rapid sea lice rejection [45].

In this work, we evaluated the familiar variability and heritability of Nkef, Tnfα, and Il-8 in skin and gills as possible molecular traits associated with resistance against sea lice *C. rogercresseyi* in Atlantic salmon. 

## 2. Materials and Methods

### 2.1. Ethics Statement

This work was carried out under the Canadian Council on Animal Care guidance for the care and use of experimental animals. The protocol was approved by the Bioethics Committee of the Pontificia Universidad Católica de Valparaíso and the Comisión Nacional de Investigación Científica y Tecnológica de Chile (FONDECYT No. 1140772). Animals were fed daily ad libitum with a commercial diet. To reduce stress during handling, vaccination was performed on fish sedated with AQUI-S (50% Isoeugenol, 17 mL/100 L water). Fish were euthanized by an overdose of anesthesia (AQUI-S, 50 mL/100 L).

### 2.2. Fish Population and Sea Lice Challenge

The animals used in this study came from an experimental challenge where 75 full-sib families of *Salmo salar* were infested with *C. rogercresseyi* as described below. Fish originated from a breeding program of Aquainnovo, Chile. Experimental families originated from 75 females who, under a nested mating scheme, mated with 40 males. Fish were pit tagged as smolts with an average body weight of 31.0 ± 8.0 g and subsequently acclimated for 3 weeks under seawater conditions (salinity of 33% and a temperature of 12 ± 1 °C). Prior to the transfer of the fish to the experimental center and as required by the authority, a group of 10 fish was analyzed to verify that they are free of pathogens in a SERNAPESCA certified laboratory. Fish health was verified by qRT-PCR against two viral and five bacterial diseases (ISAV, IPNV, *Vibrio ordalii*, *Flavobacterium psychrophilum*, *P. salmonis*, and *Renibacterium salmoninarum*).

In total, 1511 fish with an average body weight of 130 g were distributed in equal numbers per family in three tanks of 6 m^3^ each and infested with *C. rogercresseyi* larvae at the infective stage (copepodids). Copepods were obtained from ovigerous females that infected adult Atlantic salmon fish at Fundación Chile’s Aquadvise Experimental Center (Quillaipe, X region, Chile). Fundación Chile verified in a pool of larvae that they were free of pathogens by using a PCR test on ISAV and *P. salmonis.* The ovigerous females were incubated in several 2 L glass flasks with sea water at 32 ppm (filtered at 88 μm and UV disinfected). The water temperature was maintained between 13 and 14 °C in darkness and constant aeration. The infection with parasites (copepods) was carried out by concentrating the copepods in 40 L flasks and then adding them to the culture ponds. To facilitate the infestation process, the ponds were kept in the dark, with stopped flow and oxygen supplementation for ten hours, after which time the water flow was re-established [46]. Fifty thousand copepodids were used per tank, with approximately 100 parasites per fish. The temperature and oxygen saturation were controlled during this time. Approximately five days after infestation, the fish were sacrificed by an overdose of anesthesia. Small sections of gills (6 mm^2^ from lamella of the first arc) and skin (6 mm^2^ above the lateral line) without visible damage and sea lice attachment were stored in 80% ethanol. Pectoral, ventral, anal, caudal, and dorsal fins were removed and stored in 80% ethanol for estimating the sessile lice number (SL, chalimus stages) as a measure of resistance against sea lice, which represents 95% of the total number of parasites on fish [46]. Parasite counting was performed using a Nikon SMZ 800 (Nikon Inc., Melville, NY, USA) magnifying glass with a camera.

### 2.3. Family Selection and Immunological Phenotype

After the sea lice challenge, 21 families and 268 individuals covering a wide parasite load range from a low number of sessile lice (resistant fish) to a high number of sessile lice (susceptible fish) were selected to perform the immune and genetic analysis. In total, 7 families with the lowest and highest number of parasites were classified as susceptible and resistant, respectively, and 7 families with intermediate values of parasites were also included to represent a group with intermediate resistance/susceptibility. In addition, the body weight (BW) of each fish was registered. We measured abundances from undamaged skin and gills for three immune-related proteins: Nkef, Tnfα, and Il-8, according to Morales-Lange et al. [47]. Briefly, skin from the dorsolateral area without mucus (removed with a cell scraper to avoid contamination with blood and scales, according to Narváez et al. [48]), muscle tissue (removed with a scalpel), and gill samples were mechanically homogenized in lysis buffer on ice (ratio 1: 4, Tris 20 mM, NaCl 100 mM, 0.05% Triton X-100, 5 mM PMSF, 5 mM EDTA, and 0.2% protease inhibitor cocktail; Sigma-Aldrich, San Luis, MO, USA). Each homogenate was centrifuged at 14,000× *g* for 25 min at 4 °C, and the supernatant was stored at −80 °C until use. Total proteins were quantified by a Pierce BCA protein assay kit (Thermo Fisher Scientific, Waltham, MA, USA) following manufacturer’s instructions. Then, each sample was diluted in carbonate buffer (60 mM NaHCO3, pH 9.6), seeded (in duplicates) at 35 ng/µL (100 µL) in a Maxisorp plate (Nunc, Thermo Fisher Scientific), and incubated overnight at 4 °C. Thereafter, each well was blocked with 1% Bovine Serum Albumin (BSA) for 2 h at 37 °C. Plates were incubated for 90 min at 37 °C with the primary antibody anti-synthetic epitope diluted in BSA (Appendix A) and later with the second antibody HRP (Thermo Fisher Scientific) for 60 min at 37 °C in 1:7000 dilution. Finally, 100 µL per well of chromagen substrate 3,3′,5,5′-tetrame thylbenzidine (TMB) single solution (Invitrogen, Waltham, MA, USA) was added and incubated for 30 min at room temperature. The reaction was stopped with 50 µL of 1 N sulfuric acid and read at 450 nm on a VERSAmax microplate reader (Molecular Device, San José, CA, USA). Primary antibodies against Nkef, Tnfα, and Il-8 were produced according to Bethke et al., Rojas et al., and Santana et al., respectively [27,28,49] (Appendix A and Appendix A) following parameters of antigenicity (Predicting Antigenic Peptides, Universidad Complutense de Madrid), hydrophobicity (ProtScale, Hopp and Woods, ExPASy), flexibility (ProtScale, average flexibility index, ExPASy), accessibility (ProtScale,% accessible residues, ExPASy), and location in the proposed three-dimensional structure by Phyre2 (Appendix A). For validation, antibody efficiency was calculated based on the calibration curve of the antibody against the synthetic peptide used for the immunization through indirect ELISA [50], and antibody specificity was confirmed by Western blot as described before [51].

### 2.4. Genetic Variation

The heritable variation of all traits was estimated on data from 21 families by fitting the following univariate linear mixed model using ASReml 4.1 [52]:y = Xβ + Zu + e,
where y is the data recorded for the studied traits, β is the fixed effect, u is the random animal genetic effect, and e is the residual error. Each trait included the following fixed effects or covariables: (a) the tank was included as a significant fixed effect on SL, Il-8 gills, Nkef gills, Tnfα skin, and Il-8 skin; (b) body weight was included as a significant covariable on SL; (c) sessile lice was included as a significant covariable effect on Il-8 gills, Tnfα gills, and Nkef skin. The magnitude of estimated heritability was determined following the classification of Cardellino and Rovira [53], corresponding to low (0.05–0.15), moderate (0.20–0.40), high (0.45–0.60), and very high (>0.65) heritability.

### 2.5. Statistical Analysis

The association of each immunity trait with resistance traits was tested using a non-parametric Mann–Whitney test and Pearson’s correlation. Further, familiar phenotypic correlation coefficients were estimated between the six immunity traits and the number of sessile lice. Statistical analysis was performed using GraphPad Prism 8.0 software (Dotmatics, San Diego, CA, USA) and R (R Core Team, 2022).

## 3. Results

### 3.1. Sea Lice Resistance and Immunity Traits

A large variation of sea lice resistance and immunity traits were found between full sibling families, indicating considerable genetic variation, as shown in Figure 1, Figure 2 and Figure 3 and Appendix A. The seven most resistant full sib families had an average of 8.7 ± 0.9 sessile lice, while the seven most susceptible had 43.3 ± 2.0 sessile lice. Therefore, the most susceptible families had four times more sea lice than the resistant ones (Figure 1). In relation to the immunity traits, the highest coefficient of variation was found on the gills in Tnfα (50.7%), and the lowest coefficient of variation was found on the gills in Il-8 (22.6%) (Table 1). On a family level, a high and significant phenotypic correlation (r > 0.67) was found between the number of sea lice and all immunity traits, except for Il-8 on skin (Appendix A).

Regarding protein measurements, our results showed that Nkef presented a protein production dependent on the parasite load in both undamaged skin and gills tissues; thus, susceptible fish had more Nkef than resistant fish (Figure 4A,B). Additionally, higher levels of Nkef were found in the undamaged skin than in the gills of resistant and susceptible fish (*p* < 0.0001). Tnfα presented higher levels in susceptible fish than resistant fish for both undamaged skin and gills tissues. Higher levels of Tnfα were found in the skin than in the gills for resistant fish (*p* < 0.0001), although this difference was not observed in susceptible fish (Figure 4A,B). Il-8 showed a similar protein production pattern to Nkef and Tnfα. Resistant and susceptible fish did not show a significant difference in skin for Il-8 (Figure 4A). While in gills, resistant fish secreted the highest amount of Il-8 and showed significant differences (*p* < 0.0001; Figure 3B), in contrast to that observed in Nkef and Tnfα.

### 3.2. Heritability of Immunity Traits

Nkef showed the highest heritability level in skin and gills tissues (h^2^: 0.96 ± 0.14 and 0.97 ± 0.11, respectively). Tnfα showed moderate to high heritability in skin and gills (h^2^: 0.53 ± 0.17 and 0.32 ± 0.14, respectively). In contrast, the heritability estimations for Il-8 were moderate to low but not significantly different from zero in both tissues (h^2^ skin: 0.22 ± 0.12 and h2 gills: 0.09 ± 0.08, respectively; Table 1).

## 4. Discussion

In this study, we measured immunity traits as resistance indicators of Atlantic salmon against *C. rogercresseyi*. Specifically, we evaluated if Nkef, Tnfα, and Il-8 production levels in undamaged skin and gills could be associated with resistance against sea lice. Immunity traits showed moderate to very high heritability for Nkef and Tnfα, providing evidence that both could be included as selection criteria for resistance against sea lice in Atlantic salmon. Genetic variation in similar immunological traits in other fish showed low to high heritability, ranging from 0.1 to 0.58 [54,55,56,57], comparing with the results presented in this work. Thus, our results highlighted the potential of immunity traits, particularly Nkef in skin and gills and Tnfα in skin, as resistance indicators of Atlantic salmon against *C. rogercresseyi*. However, despite its higher heritability, it is more expensive to measure on a large scale when compared to the number of parasites per fish phenotype. The polygenic nature of resistance to *C. rogercresseyi* measured as the number of parasites has been previously described, with a low to moderate magnitude ranging from 0.10 to 0.32 [10,20,58,59], and genomic selection using SNP arrays has also been proposed. QTL studies and transcriptional analyses suggest several pathways by which Atlantic salmon can express resistance against *C. rogercresseyi*, including immune response, T cell regulation genes, and nutritional immunity traits [60].

Regarding protein measurements, our results showed that Nkef presented a protein production that was dependent on the parasite load in both the undamaged skin and gills tissues; thus, susceptible fish had more Nkef than resistant fish. Additionally, higher levels of Nkef were found in undamaged skin than in gills for resistant and susceptible fish, respectively. In a previous study, our group detected a significant increase in Nkef protein in the gills and skin of Atlantic salmon infected with *Caligus* compared to non-infested tissue [27]. Interestingly, in the same study, we detected Nkef in the head kidney and the spleen of infected rainbow trout, suggesting a systemic response against sea lice [27]. These results indicate that Atlantic salmon increase the production of Nkef as a response to parasite-induced oxidative stress, which has been previously described. For instance, during *L. salmonis* infection, fish induce ROS production, as evidenced by increases in the antioxidant thioredoxin [61] and the upregulation of cytokines that are responsible for stimulating ROS production [9,61,62,63]. Sea lice respond to this ROS increase by raising their own scavenge molecules such as Nkef [23]. Further, Nkef has been suggested to have a possible host resistance role during IHNV infection in Atlantic salmon [32], showing the relevance of Nkef as a resistance indicator in salmonids.

Tnfα presented higher levels in susceptible fish than resistant fish for both undamaged skin and gills tissues. Higher levels of Tnfα were found in the skin than in the gills for resistant fish, although this difference was not observed in susceptible fish. Sea lice infection is known to raise the gene expression and secretion of Tnfα in gills and skin [26,28,63]. This increase is higher in attachment skin (skin where lice attach to the host) than in undamaged skin [63]. Furthermore, classically activated macrophages are dominant in Atlantic salmon during a sea lice infection, exacerbating inflammation [63]. This oversecretion of Tnfα could contribute to the overall susceptibility of Atlantic salmon. Salmonids that are susceptible to infection respond more abruptly and in a more exacerbated manner which may increase the probability of death of the host during the fight against a pathogen, while resistant fish develop a slower and more sustained response over time maintaining the homeostasis [64].

Il-8 showed a similar protein production pattern to Nkef and Tnfα. Resistant and susceptible fish did not show a significant difference in skin Il-8. Il-8 is a chemokine that induces the migration of leukocytes to the infection site in fish [43,44], particularly neutrophils [65]. Neutrophil migration has been associated with a central role in the triggering of inflammatory processes and in the resolution of inflammation in the infection site [66]. Thus, different studies have shown that the expression of Il-8 is relatively stable in the skin where sea lice are not attached [63,67,68]. While in gills, resistant fish secreted the highest amount of Il-8 and showed significant differences compared to susceptible fish, unlike what was observed in Nkef and Tnfα. We observed a difference in the levels of IL-8 between resistant and susceptible fish families that can indicate a protective role of IL-8 in the gill. This result is the first description of Il-8 production levels in Atlantic salmon during a challenge with sea lice. A similar immune response has been observed in *Latris lineata* in a challenge with the ectoparasite *Chondracanthus goldsmidi* [69]. Infestation produces a significant increase in *il-8* gene expression in gills [69]. However, in contrast to sea lice, this ectoparasite is commonly found in gills.

Other immune parameters have been studied as traits reflecting disease resistance in fish, such as cytokine transcription, lysozyme activity, bactericidal activity, and complement activity [55,56,57,70]. From genome-wide association studies, some genes with functions in the immunologic response have been identified as resistance candidates for sea lice infestation in Atlantic salmon [71]; therefore, it would be interesting to study these genes at the protein level. For example, Correa and collaborators found a significant SNP in chromosome 21, which codes *collagen alpha-1* as an initiator of inflammatory cytokine signaling [71]. Additionally, Robledo and collaborators found two candidate genes in chromosomes 3 and 21, *tobi* and *stk17b* [10]. These genes are associated with cell proliferation and T cell function. 

## 5. Conclusions

In summary, significant familiar variation for resistance to *C. rogercresseyi* was found in this study, with moderate to very high heritability for Nkef in the skin and gills and Tnfα in the skin. Nkef and Tnfα protein expressions were related to resistance against sea lice in Atlantic salmon. Thus, resistant fish were able to elicit a moderate immune response, producing a cytokine and an effector molecule, i.e., Tnfα, Il-8, and Nkef, whereas susceptible fish showed the same response but in a higher magnitude against sea lice. Resistance against sea lice in Atlantic salmon may be associated with the ability to maintain a moderate immune response. This result is concordant with recent findings regarding Atlantic salmon infected with *C. rogercresseyi*, where susceptible fish had higher expression levels of immune molecules than resistant fish in undamaged skin [20]. Further studies are needed to investigate how this selection strategy could reduce the intensity and frequency of sea lice infections in salmon farming systems.

## Figures and Tables

**Figure 1 biology-12-01078-f001:**
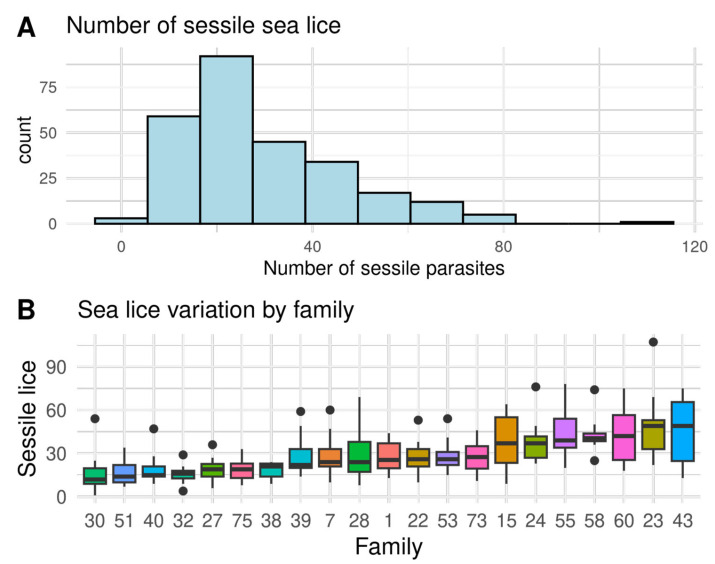
(**A**) Histogram of the total number of sessile lice (SL), and (**B**) the ranking of variation by full sibling families of Atlantic salmon from highly resistant (a low number of sea lice) to highly susceptible (a high number of sea lice). Black dots represent outliers within families.

**Figure 2 biology-12-01078-f002:**
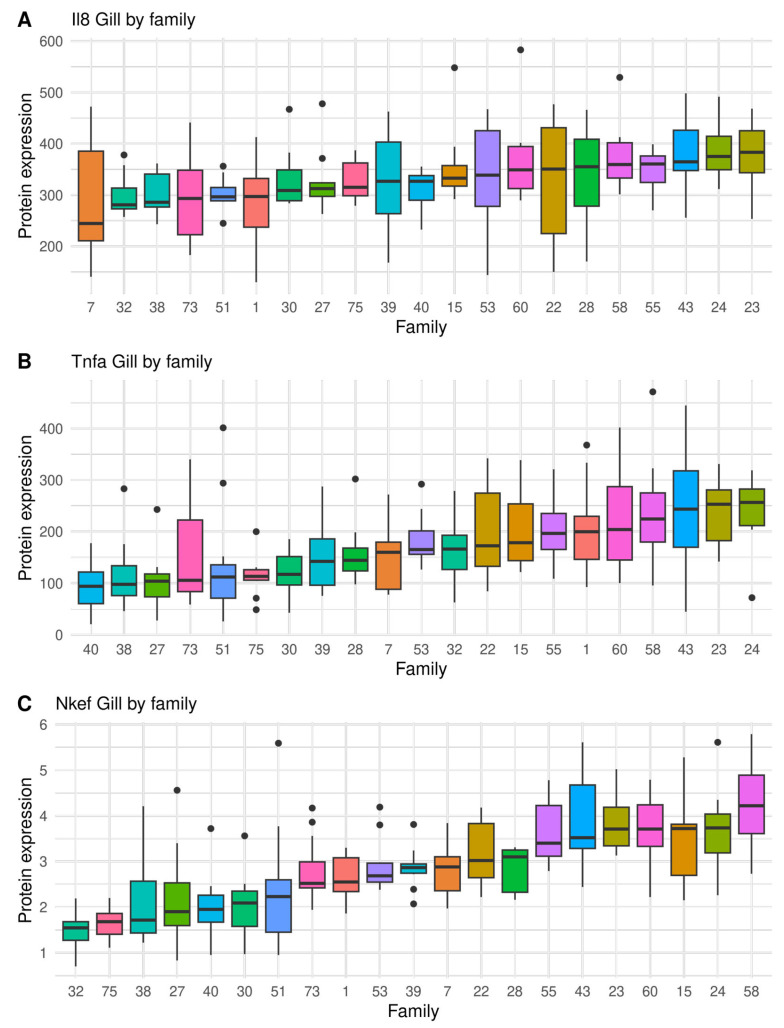
Boxplot showing full sibling family variation in immunity traits measured as protein expression (ng μL^−1^) on the gill, by indirect ELISA, in salmon infected by sea lice (chalimus sessile lice): (**A**) Il-8, (**B**) Tnfα, and (**C**) Nkef. Black dots represent outliers within families.

**Figure 3 biology-12-01078-f003:**
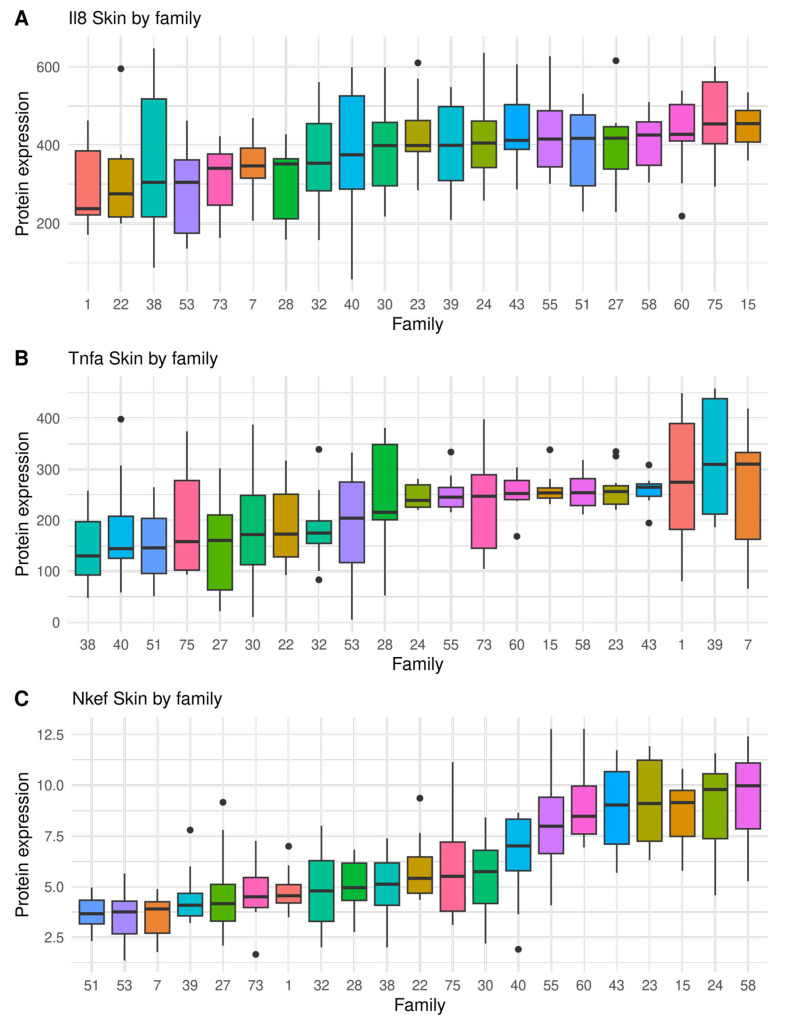
Boxplot showing full sibling family variation in immunity traits measured as protein expression (ng μL^−1^) on undamaged skin, by indirect ELISA, in salmon infected by sea lice (chalimus sessile lice): (**A**) Il-8, (**B**) Tnfα, and (**C**) Nkef. Black dots represent outliers within families.

**Figure 4 biology-12-01078-f004:**
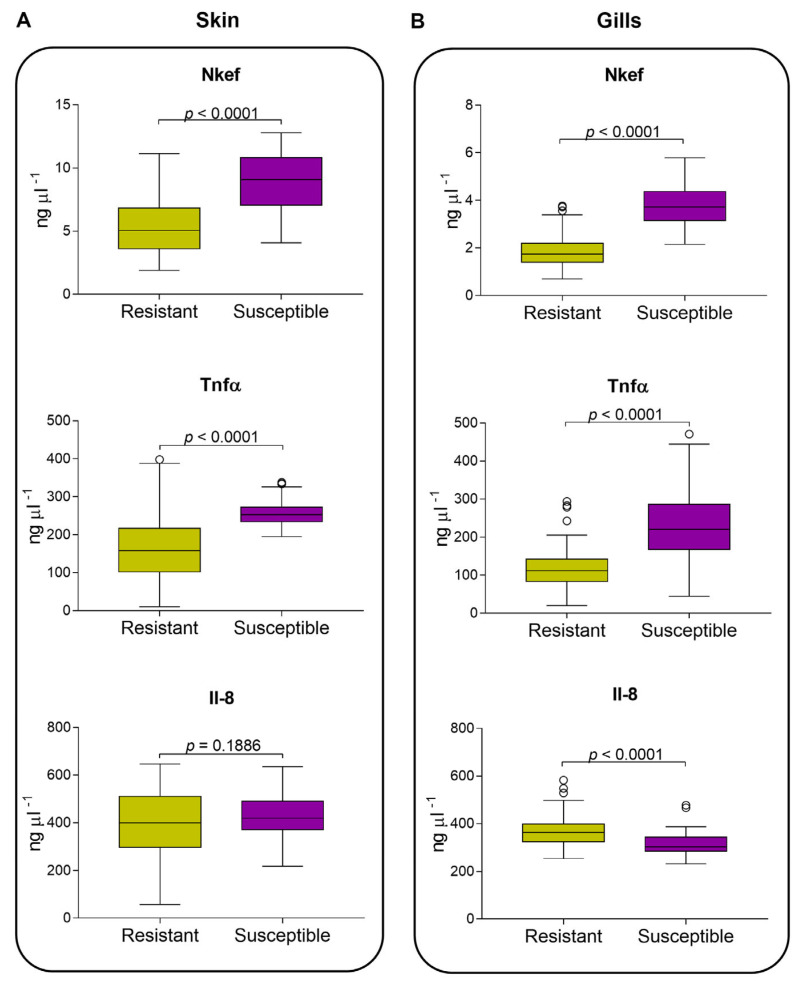
Atlantic salmon undamaged skin and gills secretion of immunity traits measured by indirect ELISA. Nkef, Tnfα, and Il-8 in undamaged skin (**A**) and gills (**B**) from seven families classified as resistant and susceptible to sea lice. Immune traits between groups were compared using a non-parametric Mann–Whitney test.

**Table 1 biology-12-01078-t001:** Descriptive statistics and heritability estimates of immunity and parasite load traits. Immunity traits measured by indirect ELISA (ng μL^−1^) and sea lice load (SL) in Atlantic salmon were evaluated on undamaged skin and gills. Bold indicates significantly non-zero heritability. Abbreviations: SL, the number of sessile lice; na, not applicable; and ns, non-significant.

Trait	Tissue	N	Mean	SD	CV	h^2^ (SE)	h^2^ Levels
Nkef	Skin	229	6.2	2.7	43.7	**0.96 ± 0.14**	Very high
Tnfα	Skin	227	221.6	91.0	41.0	**0.53 ± 0.17**	High
Il-8	Skin	214	384.6	120.1	31.2	0.22 ± 0.12 ns	Moderate
Nkef	Gills	231	2.9	1.1	37.3	**0.97 ± 0.11**	Very high
Tnfα	Gills	229	174.3	88.4	50.7	**0.32 ± 0.14**	Moderate
Il-8	Gills	238	332.3	75.1	22.6	0.09 ± 0.08 ns	Low
SL	Skin	267	29.1	16.4	56.5	**0.58 ± 0.17**	High

## Data Availability

The raw data and analysis supporting the conclusions of this article will be made available by the authors through Github, a public repository, without undue reservation. https://github.com/GenomicsLaboratory/ReproducibleResearch, accessed on 10 June 2023.

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
