# Peer review of "Heritability of Immunity Traits and Resistance of Atlantic Salmon against the Sea Louse Caligus rogercresseyi"

_biology, 2023, doi:10.3390/biology12081078_

Round 1

Reviewer 1 Report

I reviewed the manuscript entitled “Heritability of immunity traits and resistance of Atlantic salmon against the sea louse Caligus rogercresseyi”. It is a well structured work, which analyze the production of three proteins in salmon infected with sea lice. Some comments, which could be considered before publication, are provided below.   

The manuscript version of this reviewer has not line numbers, which impairs a smooth revision.  

Intro

Line 1. Maybe it is necessary to clarify what sea lice are? Copepods of the family Caligidae, mainly represented by two genera Lepeophtheirus and Caligus.

Line 2. Consider “The harmful effect of these parasites…”; note that the correct name is Caligidae, maybe you want to say “caligid parasites”. Check all the document.

Line 6. Bath treatments generally use antiparasitic compounds, so it is confusing to state antiparasitic as method. I would say that all methods pretend to be antiparasitic.

Vibrio, Tenacibaculum, and Aeromonas ... … Caligus…  in italics? Caligus is a genus name

Materials and methods

Second paragraph

Sessile stage? I am not sure that you infected with sessile stage, maybe you want to say infective stage.

How copepodids were obtained for challenge? How copepodids were put into the tank?

2nd paragraph, line 4. After infection? How the infection was determined? Maybe you are referring after the moment the copepodids were put into the tank.

Regarding Table 1 and Figure 1, I am not sure if sessile lice is a correct term. It can be correct, but I think that it is necessary first to provide an brief explanation about the copepod development; then the term sessile (or chalimus) may be more understandable.

Author Response

Dear Reviewer,

We are deeply grateful for your constructive contributions, suggestions, and comments, which have helped us greatly to improve the quality of the paper. Please find below responses to each of your observations.

Comments and Suggestions for Authors

I reviewed the manuscript entitled “Heritability of immunity traits and resistance of Atlantic salmon against the sea louse Caligus rogercresseyi”. It is a well structured work, which analyze the production of three proteins in salmon infected with sea lice. Some comments, which could be considered before publication, are provided below.   

The manuscript version of this reviewer has not line numbers, which impairs a smooth revision.  

 Introduction

Question: Line 1. Maybe it is necessary to clarify what sea lice are? Copepods of the family Caligidae, mainly represented by two genera Lepeophtheirus and Caligus.

Reply: Thank you for your comments. We added these sentences “Sea lice are fish parasitic copepods of the family Caligidae, mainly represented by two genera Caligus and Lepeophtheirus. Two of the principal ectoparasite of these genera affecting salmon farming are Caligus rogercresseyi in the southern hemisphere and Lepeophtheirus salmonids in the northern hemisphere”

Question: Line 2. Consider “The harmful effect of these parasites…”; note that the correct name is Caligidae, maybe you want to say “caligid parasites”. Check all the document.

Reply: Thank you for your comments. We changed it throughout the manuscript.

Question: Line 6. Bath treatments generally use antiparasitic compounds, so it is confusing to state antiparasitic as method. I would say that all methods pretend to be antiparasitic.

Reply: Thanks for the clarification, it has been changed in the manuscript.

Question: Vibrio, Tenacibaculum, and Aeromonas ... … Caligus…  in italics? Caligus is a genus name

Reply: Thank you for your comments. We added italics to all mentioned names.

Materials and methods

Question: Second paragraph. Sessile stage? I am not sure that you infected with sessile stage, maybe you want to say infective stage.

Reply: Thanks for the suggestion, it has been changed in the text to “ infective stage”.

Question: How copepodids were obtained for challenge? How copepodids were put into the tank?

Reply: We added this information in the materials and methods section with the following paragraph: “Copepods were obtained from ovigerous females that infected adult Atlantic salmon fish at Fundación Chile's Aquadvise Experimental Center (Quillaipe, X region, Chile). Fundación Chile verified in a pool of larvae that they were free of pathogens using a PCR test on ISAV and P. salmonis. The ovigerous females were incubated in several 2 L glass flasks with sea water at 32 ppm (filtered at 88 mi-crons and UV disinfected). The water temperature was maintained between 13 and 14ºC in darkness and constant aeration. The infection with parasites (copepods) was carried out by concentrating the copepods in 40-liter flasks and then adding them to the culture ponds. To facilitate the infestation process, the ponds were kept in the dark, with stopped flow and oxygen supplementation for ten hours, after which time the water flow was reestablished.”.

Question: 2nd paragraph, line 4. After infection? How the infection was determined? Maybe you are referring after the moment the copepodids were put into the tank.

Reply: Yes, we mean to the moment the copepodids were put into the tank.

Question: Regardingg Table 1 and Figure 1, I am not sure if sessile lice is a correct term. It can be correct, but I think that it is necessary first to provide an brief explanation about the copepod development; then the term sessile (or chalimus) may be more understandable.

Reply: We included new information in the introduction about the developmental stages of C. rogercresseyiC. rogercresseyi has eight developmental stages, these include two free-swimming nauplius, a copepodid stage that infests and attaches to the host, four attached chalimus, and finally a mobile adult stage [1]. Usually, newly attached copepotid stages and chalimus stages are known as sessile parasites in contrast to the adult that is capable of moving on the host.”

Also, we included in material and methods the meaning of sessile lice “for estimating the sessile lice number (SL, chalimus stages)”.

Reviewer 2 Report

The authors present interesting information about the heritability of immunity traits and resistance of Atlantic salmon against the sea lice Caligus rogercresseyi.  The manuscript is well presented with appropriate figures and tables. My main comment is about the discussion section. The article title refers to the heritability of the immunity traits, however, there is only one mention of heritability in the discussion in the first paragraph. Therefore, I encourage the authors to dive deeper into the heritability estimates of each parameter and what it could mean for breeding selection programs. Could these markers be used as selection tools to breed for infestation resistance? Specific comments are in the attached file.

Minor editing of the English is required. I have made comments on the attached document.

Author Response

Dear Reviewer,

We are deeply grateful for your constructive contributions, suggestions, and comments, which have helped us greatly to improve the quality of the paper. Please find below responses to each of your observations.

Comments and Suggestions for Authors

The authors present interesting information about the heritability of immunity traits and resistance of Atlantic salmon against the sea lice Caligus rogercresseyi.  The manuscript is well presented with appropriate figures and tables. My main comment is about the discussion section. The article title refers to the heritability of the immunity traits, however, there is only one mention of heritability in the discussion in the first paragraph. Therefore, I encourage the authors to dive deeper into the heritability estimates of each parameter and what it could mean for breeding selection programs. Could these markers be used as selection tools to breed for infestation resistance? Specific comments are in the attached file.

Introduction

Suggestion: Vibrio, Tenacibaculum, and Aeromonas Italics

Reply: Thanks for the suggestion, we added italics to these names.

Suggestion: caligus in italics in all the text

Reply: Thanks for the suggestion, we changed these in all the manuscript.

Suggestion: “leads” to “leading”

Reply: Thanks for the suggestion, it has been changed from “leads” to “leading”

Suggestion: “is member” to “is a member”

Reply: Thanks for the suggestion, it has been changed from “is member” to “is a member”

Suggestion: “play” to “plays”

Reply: Thanks for the suggestion, it has been changed from “play” to “plays”

Suggestion: “antioxidant activity” to “antioxidant activity role”

Reply: Thanks for the suggestion, it has been changed from “antioxidant activity” to “antioxidant activity role”

Materials and Methods

Question: how was this health check performed? what tissue was sampled for this? how were the fish sampled?please add a brief description of this methods that are based on this regulation.

Reply: The fish for the experiment were provided by the company aquainnovo, who by law must certify that the fish do not have pathogens before the movement of fish to the experimental center. For this, there are authorized laboratories in Chile. We do not know which laboratory performed the analysis, so we cannot give more details about the test performed.

We clarified this nformation in the Materials and Methods section “Prior to the transfer of the fish to the experimental center and as required by the authority, a group of 10 fish was analyzed to verify that they are free of pathogens in a SERNAPESCA certified laboratory. Fish health was verified by qRT-PCR against two viral or five bacterial diseases (ISAV, IPNV, Vi”brio ordalii, Flavobacterium psychrophilum, P. salmonis, and Renibacterium salmoninarum).

Question: were there any control tanks in which fish were not exposed to the sea lice infestation?

Reply: No parasite-free tanks were considered in this study, because it was not of interest to assess the response of fish to infection, but rather to compare the difference between resistant and susceptible fish in some biomarkers previously identified as relevant in the immune response.

Question: is this the same dose as mentioned in the section above? If so, use the same units to present the same information.

Reply: Yes, it is the same information we changed the information to maintain the same units.

Question: give an estimation of how small were the samples, either in mg or as size (e.g., 1x1cm)

Reply: We added this information to the manuscript “Small sections of gills (6 mm2 from lamella of the first arc) and skin (6 mm2 above the lateral line) without visible damage and sea lice attachment were stored in 80% ethanol”

Suggestion: “a stores” to “were stored”

Reply: Thanks for the suggestion, it has been changed from “a stores” to “were stored”

Suggestion: “represent” to “represents”

Reply: Thanks for the suggestion, it has been changed from “represent” to “represents”

Question: Table S1?

Reply: Yes, is Table S1. We changed the text.

Question: Table S1 and Figure S1?

Reply: Yes, is Table S1 and Figure S1. We changed the text.

Suggestion: “an” to “and”

Reply: Thanks for the suggestion, it has been changed from “an” to “and”

Results

Question: This is results and therefore it should be in the results section.

Reply: This section was duplicate, we eliminated this sentence in this place.

Discussion

Question: The article title refers to the heritability of the immunity traits. However, there is only one mention of heritability in the discussion in the first paragraph. Therefore, I encourage the authors to dive deeper into the heritability estimates of each parameter and what it could mean for breeding selection programs. Could these markers be used as selection tools to breed for infestation resistance?

Reply:  in the first paragraph of discussion we delve into this matter. “Genetic variation in similar immunological traits in other fish showed low to high her-itability ranging from 0.1 to 0.58 [56-59], comparing with the results presented in this work. Thus, our results highlighted the potential of immunity traits, particularly Nkef in skin and gills, and Tnfα in skin as resistance indicators of Atlantic salmon against C. rogercresseyi. However, despite its higher heritability, it is more expensive to measure on a large scale when compared to the number of parasites per fish phenotype. The polygenic nature of resistance to C. rogercresseyi measured as number of parasites has been previously described, with low to moderate magnitude ranging from 0.10 to 0.32 [10,21,60,61] and genomic selection using SNP arrays also has been proposed. QTL studies and transcriptional analyzes suggest several pathways by which Atlantic salm-on can express resistance against C. rogercresseyi including immune response, T cell regulation genes (60) and nutritional immunity traits [62]. “

Suggestion: “measurements” to “measure”

Reply: Thanks for the suggestion, it has been changed from “measurements” to “measure”

Question: “during Lepeophtheirus salmonis northern sea lice infection fish host induces ROS production” this sentence doesn't read well. Do you mean that the host (fish) induces ROS production? Please revise this sentence.

Reply: Thanks for your comment. Yes, me mean that Host produce ROS. We rephrase the sentence to “during Lepeophtheirus salmonis infection fish induces ROS production”

Suggestion: “responding” to “responds”

Reply: Thanks for the suggestion, it has been changed from “responding” to “responds”

Suggestion: “raising” to “by raising”

Reply: Thanks for the suggestion, it has been changed from “raising” to “by raising”

Suggestion: “suggested” to “suggested to have”

Reply: Thanks for the suggestion, it has been changed from “suggested” to “suggested to have”

Suggestion: “attachment is frecuently observed” to “attach to the host”

Reply: Thanks for the suggestion, it has been changed from “attachment is frecuently observed” to “attach to the host”

Question: showing significant differences compared to resistant fish?

Reply: We changed the sentence to “While in gills, resistant fish secreted the highest amount of Il-8 showed significant differences compared to susceptible fish, unlike what was observed in Nkef and Tnfα”

Question: “Il-8 is a chemokine that induces the migration of leukocytes to the infection site”. please delve into this a little. Does this migration of leukocytes help fighting the infection directly or has another mechanism of action?

Reply: Thanks for the commentary we complete the sentence to “Il-8 is a chemokine that induces the migration of leukocytes to the infection site in fish [39,40], particularly neutrophils (Havixbeck, 2015). Neutrophil migration has been associated with a central role in the triggering of inflammatory processes and in the resolution of inflammation in the infection site (Havixbeck, 2016). Even though, different studies have been shown that the expression of Il-8 is relatively stable in the skin where sea lice are not attached [56,58,59]. We observed a difference in the levels of IL-8 between resistant and susceptible fish families that can indicate a protective role of IL-8 in the skin”

Suggestion: “observed” to “observed in”

Reply: Thanks for the suggestion, it has been changed from “observed” to “observed in”

Suggestion: “ a difference of sea lice” to “in contrast to sea lice”

Reply: Thanks for the suggestion, it has been changed from “ a difference of sea lice” to “in contrast to sea lice”

Suggestion: “study” to “study these genes”

Reply: Thanks for the suggestion, it has been changed from “study” to “study these genes”

References

Suggestion: Make sure all genus and species names are italicized. Also, revise all journal names as some are abbreviated while others are not.

Reply: Thanks for your suggestion. We revised all the references.

Reviewer 3 Report

The work by Torrealba et al. is of particular interest to the salmon culture sector and needs some improvement before publication. There are items from the Materials and Methods that are absent. The authors mention six different RT-PCR assays but provide no Materials or Methods or references to them. Additionally, the manner in which the different families were selected for the second round of analyses is poorly described; there is only a brief mention of the fact that pit tags were used. Also, why was skin without mucus used for analysis; and how was mucus removed? The Figures need to be improved. Figure 1a is exemplary for its lack of labeling of the y-axis. Also no explaination of the black dots in the box plots. All figures need to be improved in terms of their captions. There are some issues with English grammar and syntax that aslo need correcting.  

Grammar and syntax and also in a few places spelling need correction. You shouldn´t start a sentence with "However, although....); either one or the other but not both. Mating (to mate) is spelled with one "t". Proper use of prepositions is somewhat mixed. The overall sense of the authors intent is there but reading does not "flow" due to some of these small problems.

Author Response

Dear Reviewer,

We are deeply grateful for your constructive contributions, suggestions, and comments, which have helped us greatly to improve the quality of the paper. Please find below responses to each of your observations.

Comments and Suggestions for Authors

The work by Torrealba et al. is of particular interest to the salmon culture sector and needs some improvement before publication.

Question: There are items from the Materials and Methods that are absent. The authors mention six different RT-PCR assays but provide no Materials or Methods or references to them.

Reply: The fish for the experiment were provided by the company aquainnovo, who by law must certify that the fish do not have pathogens before the movement of fish to the experimental center. For this, there are authorized laboratories in Chile. We do not know which laboratory performed the analysis, so we cannot give more details about the test performed.

We clarified this information in the Materials and Methods section “Prior to the transfer of the fish to the experimental center and as required by the authority, a group of 10 fish was analyzed to verify that they are free of pathogens in a SERNAPESCA certified laboratory. Fish health was verified by qRT-PCR against two viral or five bacterial diseases (ISAV, IPNV, Vibrio ordalii, Flavobacterium psychrophilum, P. salmonis, and Renibacterium salmoninarum).

Question: Additionally, the manner in which the different families were selected for the second round of analyses is poorly described; there is only a brief mention of the fact that pit tags were used.

Reply: We clarified this nformation in a new section entitled 2.3. Family selection and  immunological phenotype. “After the sea lice challenge, 21 families and 268 individuals covering a wide parasite load range from low number of sessile lice (resistant fish) to a high number of sessile lice (susceptible fish) were selected to perform the immune and genetic analysis. In total, 7 families with the lowest and highest number of parasites were classified as susceptible and resistant, respectively. While the 7 families with intermediate values ​​of parasites were also included to represent a group with intermediate resistance / susceptibility.”

Question: Also, why was skin without mucus used for analysis; and how was mucus removed?

Reply: In our study we have focused on the skin and not on the mucus because we were interested in the evaluation of the biomarkers directly related to the organ and not in the secretion. The skin has mucosa-associated lymphoid tissues (Salinas, 2015, 10.3390/biology4030525) with cells that can be modulated and coordinated by cytokines (e.g., TNFa and IL8, related to type 1 response) and NKEF for the reinforcement of the cytotoxic process mediated by innate immune cells. In addition, we did not perform the evaluation of parameters in the mucus since we considered the report by Djordjevic et al. (2021, 10.1111/anu.13248 ) where they made a proteomic description of the skin and gut mucus of Atlantic Salmon, which showed a composition highly related to enzymes, mucin-like proteins, immunoglobulins and other effector molecules, and not so much in protein coordinators such as cytokines.

Suggestion: The Figures need to be improved.

Reply: Thanks for the suggestion. We improved the figures.

Question: Figure 1a is exemplary for its lack of labeling of the y-axis. Also no explanation of the black dots in the box plots. All figures need to be improved in terms of their captions.

Reply: Labels y-axis and captions for figure 1, 2 and 3 were improved.

Suggestion: There are some issues with English grammar and syntax that also need correcting.  

Reply: Thanks for the suggestion. We improved the English grammar in the manuscript.

Comments on the Quality of English Language

Question: Grammar and syntax and also in a few places spelling need correction. You shouldn´t start a sentence with "However, although....); either one or the other but not both. Mating (to mate) is spelled with one "t". Proper use of prepositions is somewhat mixed. The overall sense of the authors intent is there but reading does not "flow" due to some of these small problems.

Reply: Thank you for the commentary. We changed the sentence “However, the detection….” Also, we changed the word “matting” to “mating”. We improved the English grammar in all the manuscript.
